# Recent Findings in Onychomycosis and Their Application for Appropriate Treatment

**DOI:** 10.3390/jof5010020

**Published:** 2019-02-22

**Authors:** Michel Monod, Bruno Méhul

**Affiliations:** 1Service de Dermatologie, Laboratoire de Mycologie, Centre Hospitalier Universitaire Vaudois (CHUV), 1011 Lausanne, Switzerland; 2Galderma R&D, 06902 Sophia-Antipolis, France; bruno.mehul@syneoshealth.com

**Keywords:** onychomycosis, *Trichophyton*, *Fusarium*, *Acremonium*, *Aspergillus*, terbinafine, itraconazole, amphotericin B, antifungal drug resistance

## Abstract

Onychomycosis is mainly caused by two dermatophyte species, *Trichophyton rubrum* and *Trichophyton interdigitale*. A study of nail invasion mechanisms revealed that the secreted subtilisin Sub6, which has never been detected under in vitro growth conditions, was the main protease secreted by *T. rubrum* and *T. interdigitale* during infection. In contrast, most of the proteases secreted during the digestion of keratin in vitro were not detected in infected nails. The hypothesis that proteases isolated from dermatophytes grown in a keratin medium are virulence factors is no longer supported. Non-dermatophyte fungi can also be infectious agents in nails. It is necessary to identify the infectious fungus in onychomycosis to prescribe adequate treatment, as moulds such as *Fusarium* spp. and *Aspergillus* spp. are insensitive to standard treatments with terbinafine or itraconazole, which are usually applied for dermatophytes. In these refractory cases, topical amphotericin B treatment has shown to be effective. Terbinafine treatment failure against dermatophytes is also possible, and is usually due to resistance caused by a missense mutation in the squalene epoxidase enzyme targeted by the drug. *Trichophyton* resistance to terbinafine treatment is an emerging problem, and a switch to azole-based treatment may be necessary to cure such cases of onychomycosis.

## 1. Introduction

Two species of dermatophyte, *Trichophyton rubrum* and *Trichophyton interdigitale*, are the cause of most onychomycosis, often following fungal infection of the interdigital or plantar spaces [1]. However, the infectious agents of nail infections are not only dermatophytes but also moulds and yeasts of the genus *Candida*. Dermatophyte onychomycosis is also called tinea unguium. The diagnosis of onychomycosis is usually primarily based on the detection of a fungus in nail samples by direct mycological examination, then on the isolation and identification of the infectious fungus in culture [1]. The purpose of this article is to briefly review recent findings on fungal nail infection and to present new perspectives for improving their treatment. Tools for the diagnosis of onychomycosis and rapid identification of the pathogen, which remain a challenge for the clinician, are summarized in Table 1.

## 2. Direct Mycological Examination of Nail Samples

Various skin diseases look like but are not mycoses. Before any drugs are prescribed, direct mycological examination of skin, nail, or hair samples is an essential step for confirming the clinical diagnosis of fungal infection in dermatology. Many laboratories still use 20%–30% KOH solution to dissociate dermatological samples [1]. Another effective solution for the dissociation of keratinized samples is a 1:1 mixture of water and ethanol containing 10% Na_2_S [2].

It is difficult to detect fungal elements using light microscopy without contrast. Direct microscopical examination using fluorochromes, which specifically bind to vegetal and fungal cell wall polysaccharides, is by far the most sensitive contrasting technique for detecting hyphae and spores in dermatological samples [2,3,4,5]. When the preparation is illuminated by light with a wavelength between 360 and 440 nm, the hyphae and spores become fluorescent and are immediately detected (Figure 1). Fluorescence microscopy allows for better observation of fungal morphology. A recent improvement has been the use of fluorescence microscopy with an LED light source, which intensifies the fluorescence. In addition, LED lamps have a long service life and can be switched on without having to wait for the lamp to cool down.

## 3. Fungal Cultures from Nail Samples

The fungus is isolated after nail fragments are deposited on a nutrient agar medium. A commonly used medium is Sabouraud supplemented with chloramphenicol to inhibit bacterial growth [1]. It is also imperative to selectively promote the growth of dermatophytes while inhibiting the growth of associated moulds, which generally grow relatively faster. Therefore, two cultures are generally carried out in parallel, one with and the other without actidione in the medium [1]. This compound inhibits the growth of most moulds, but not dermatophytes which are not sensitive to it. Yeasts develop in 24–48 h, whereas non-dermatophyte filamentous moulds (NDMs) such as *Aspergillus* spp., *Penicillium* spp., and *Fusarium* spp., which can be infectious or transient contaminants and the most frequent species of contaminating saprophytes (*Alternaria* spp. and *Curvularia* spp.), develop in 3 to 5 days. Between 7 and 14 days are necessary for the growth and identification of dermatophytes.

## 4. Identification of Fungi In Situ in Nail Samples

Two problems arise when identifying fungi in dermatological samples when growing cultures. (i) Often, the seeded tubes (or Petri dishes) remain sterile. It has been shown that no fungi grow in about 35% of cases following a positive direct mycological examination [19]. (ii) Many NDMs are isolated. It is frequently difficult to determine whether a mould is in fact the infectious agent causing onychomycosis or a contamination. Using cultures, only repeated isolations of the same NDM indicate its involvement in nail infection with some certainty [6,20].

Conventional PCR and real-time PCR techniques are now reliable and suitable for performing the in situ identification of dermatophytes, yeasts, and NDMs in onychomycosis, provided that enough nail material is collected by the clinician [21]. Several methods aim to detect one particular species with specific primers [7,8,9]. Others aim to detect dermatophytes as well as NDMs and yeasts as infectious agents in nail samples using pan-fungal primers [10,11]. The results of fungal identification obtained by PCR in onychomycosis are representative of the infection at the time when the nail sample was collected. PCR has significantly improved the diagnosis and treatment of onychomycosis. (i) NDMs such as *Fusarium* spp., *Acremonium* spp., or *Scopulariopsis* spp. and yeasts have been identified as infectious agents of onychomycosis and not as transient contaminants with certainty [10,11]. (ii) An infectious agent can be identified in at least 70% of cases where a direct mycological examination of the nails shows fungal elements, while negative results are obtained by fungal culture [10,11]. Obtaining results within 48 hours, whereas those from fungal cultures can take at least one week, is another advantage. PCR identification of fungi in situ has shown a high prevalence of moulds in onychomycosis. The prevalence of NDMs in onychomycosis has been estimated at approximately 20% of cases, not including mixed infections [10,11]. The statistics provided on this subject from culture results vary widely in the literature from 3% to 80% [22,23].

Immunological methods are another trend in the detection of onychomycosis. An anti-dermatophyte monoclonal antibody that reacts with the cell wall polysaccharide antigen of dermatophytes was recently produced [14]. The nature of the recognized antigen was not identified, but the epitope of this antibody may be a carbohydrate moiety similar to the galactomannan or galactofuran of *Aspergillus* detected by monoclonal antibodies in cases of invasive aspergillosis in immunocompromised patients. Using this anti-dermatophyte monoclonal antibody and immunochromatography, test strips were developed and validated [15,16,17,18]. Test strips are now available as new devices to detect *Trichophyton* in infected nails (dermatophyte test Strip, JNC Corporation, Tokyo, Japan; FungiCheck, HFL Laboratories, The Netherlands). Available test strips are dedicated to dermatophyte detection in dermatological samples. However, the monoclonal antibody also reacts with *Aspergillus flavus* and *Fusarium solani* [18]. While the latter are nail infectious agents, there are no data on their possible detection in dystrophic nails.

## 5. Mechanisms of Nail Invasion

The fungus can enter the nail through the distal subungual area and the lateral nail groove, through the dorsal surface of the nail plate producing superficial onychomycosis, or through the under-surface of the proximal fold of the nail [1]. Infection may also be secondary to paronychia. Dermatophyte infections are often the result of an intertrigo, while repetitive strain injuries to the nail, especially hallux, may play a major role in mould invasion.

Research on mechanisms of nail invasion, as well as that of other keratinized tissues, primarily focuses on secreted proteases. All dermatophytes grow well in a medium containing hard keratin as the sole source of carbon and nitrogen, and most secreted proteins in culture supernatant are proteases. At alkaline pH, two subtilisins (Sub3 and Sub4) and metalloproteases of the fungalysin family (M36 family in the MEROPS database) are the major endoproteases secreted by dermatophytes, together with exopeptidases. The latter comprises two leucine aminopeptidases of the M28 family (Lap1 and Lap2) and two dipeptidyl-peptidases of the S9 family (DppIV and DppV) [24]. At acidic pH, dermatophytes were found to secrete an aspartic protease of the pepsin family (Pep1) as an endoprotease and tripeptidyl peptidases of the sedolisin family, X-prolyl peptidases and carboxypeptidases of the S10 family as exoproteases [25]. Proteases secreted in vitro were considered as virulence factors. However, proteomic analysis of proteins extracted from infected nail beds of patients with onychomycosis showed that particular subtilisin (Sub6), not detected in vitro, was the major secreted protease during infection [12,13]. Additional secreted proteins were identified, among which was the closely related Sub7 and the dipeptidyl peptidase DppV. Surprisingly, most proteases secreted in vitro during keratin digestion were not detected during infection establishment [12,13]. These results were in accordance with a transcriptomic study performed with RNA from guinea pigs infected by the dermatophyte species *Trichophyton benhamiae*, which naturally infects guinea pigs [26]. RNA sequencing analysis revealed that *SUB6* was the major gene expressed by the fungus during infection. In contrast, the genes encoding most of the major proteases secreted by the fungus in vitro were not found to be expressed in vivo, which confirmed the results of proteomic analyses with clinical samples [12,13]. Therefore, the previous hypothesis that proteases isolated from dermatophytes grown in vitro in a keratin medium are virulence factors and play a major role during nail and skin infection should be abandoned. In summary, *T. rubrum* does not grow in the nails like a saprophyte, destroying dead material, but grows parasitically like *T. benhamiae* in skin infection, attesting to the control of its transcriptome by keratinocytes.

Sub6 and DppV were first identified and characterised as major allergens of *T. rubrum*, causing immediate hypersensitivity and delayed-type hypersensitivity skin reactions [27]. Sub6 and DppV antigens have been called Tri r 2 and Tri r 4, respectively. *Trichophyton rubrum* infections cause severe asthma as a secondary immune reaction [28,29]. There is a strong association between protein sensitisation of the fungus and the severity of asthma. *Trichophyton* asthma can be controlled by systemic antifungal therapy [30,31]. *Trichophyton* asthma is not very common but is possibly underestimated due to the lack of fungal extracts for intradermal skin tests to demonstrate sensitisation.

Sub6 was revealed to be a very specific biomarker of *T. rubrum* and *T. interdigitale* [12], and monoclonal antibodies targeting *T. rubrum* and *T. interdigitale* Sub6 were developed for the detection of dermatophytes in infected nails [13]. They allowed the detection of a 40-kDa band corresponding to the full-length subtilisin and a 20 kDa band corresponding to a major degradation product of the enzyme. These antibodies were used to identify dermatophyte onychomycosis by Western blotting and ELISA with protein extracts from infected nails. As a perspective, Sub6 monoclonal antibodies could be used to develop a new strip test for the rapid diagnosis of tinea unguium.

## 6. Resistance of Onychomycoses to Standard Treatment

### 6.1. Antifungal Susceptibility Testing

The insensitivity of onychomycosis to standard antifungal therapy often leads to the request of available antifungal resistance tests. While antifungal susceptibility testing is repeatedly requested for *Candida albicans* and *Aspergillus fumigatus*, the absence or low sporulation of *T. rubrum* has always been a major complication and a limiting factor to perform this analysis in routine. Guidelines and the literature both list different media (e.g., SDA diluted 1/10, potato dextrose agar (PDA) and oatmeal agar (OTA)) on which *T. rubrum* is supposed to sporulate easily [32,33]. However, variability between strains remains present in these media with respect to quantitative sporulation. On the other hand, Chin and Knight reported in 1957 that sporulation of *T. rubrum* under increased CO_2_ tension [34], and a combination of high CO_2_ tensions and incubation on PDA growth medium proved to be optimal to obtain spores of this dermatophyte [35]. The availability of large quantities of spores subsequently used as colony forming units allows the preparation of standardised inocula to perform microdilution, checkerboard, disc diffusion, and Etest assays in the same way as for bacteria, yeasts, and *Aspergillus* spp. CSLI guidelines specify the use of RPMI 1640 medium for antifungal susceptibility tests [33]. However, SDA medium was found to be more advisable than RPMI medium when working with *T. rubrum* because this species grows poorly on this medium in contrast to yeasts and *Aspergillus* spp. [35].

### 6.2. Resistance of Dermatophytes to Standard Treatments

Terbinafine and azole are used extensively in therapy for dermatophyte infections. Terbinafine inhibits the squalene epoxidase (SQLE) involved in an early step of ergosterol biosynthesis [36], a compound essential for the structure and function of the plasma membrane in fungi. The resulting intracellular accumulation of squalene is toxic to fungal cells [37]. Azoles such as itraconazole and voriconazole act downstream of the squalene epoxidase reaction in membrane ergosterol synthesis. These cytostatic antifungal drugs inhibit the lanosterol 14-α-demethylase encoded by *ERG11* in *C. albicans* and *CYP51* in filamentous fungi [38]. Lanosterol 14-α-demethylase inhibition leads to the accumulation of sterol precursors. A defect in the synthesis of ergosterol results in the alteration of plasma membrane structure and function.

Increased exposure to antifungal drugs favours the generation of resistant microbes, and following this trend, dermatophytes resistant to terbinafine have recently emerged in several countries [39,40,41,42]. The prevalence of terbinafine-resistant *Trichophyton rubrum* nail isolates was found to be 1.0% in Switzerland (26 cases among 2056 tested isolates) [39] and unpublished work: Monod, M. et al. manuscript in preparation. The DNA sequence of the gene encoding SQLE in 17 *T. rubrum* clinical isolates from tinea pedis and tinea unguium in Switzerland with reduced terbinafine susceptibility revealed single-point mutations, leading to amino acid substitutions at one of the four amino acid positions (Leu^393^, Phe^397^, Phe^415^ and His^440^) within the SQLE protein. Four of the 17 *T. rubrum* isolates harboured the same Leu393Phe mutation previously identified in one clinical isolate by Osborne et al. [43], and four other isolates harboured the same Phe397Leu mutation previously identified in another clinical isolate by the same author [44]. These two mutations were detected in terbinafine-resistant *T. mentagrophytes* causing tinea cruris in India, attesting to the prevalence of these mutations in resistant dermatophyte clinical isolates [42]. Introduction of the detected point mutations into terbinafine-sensitive *T. mentagrophytes* generated terbinafine-resistant phenotypes for which the terbinafine MIC was comparable to that of corresponding clinical strains [39]. These results showed that terbinafine resistance in *Trichophyton* clinical isolates had to be imputed to the detected amino acid substitution in the SQLE protein. *Trichophyton* resistance to terbinafine treatment is an emerging problem, and a switch to azole-based treatment may be necessary to cure such cases of onychomycosis.

In addition to missense mutations in the genes encoding the drug target, antifungal resistance can also be mediated by overexpression of genes encoding multidrug transporters (MDR) in many fungi such as *C. albicans* and *A. fumigatus* [45,46]. Two dermatophyte ATP-binding-cassette (ABC) transporter genes, called *MDR1* and *MDR2* (for multidrug resistance 1 and 2, respectively), have been reported [47,48]. Although increased expression levels of *MDR1* and *MDR2* were observed in response to stress caused by sub-inhibitory levels of various antifungal compounds, nothing is known about the ability of both transporters to function as efflux pumps for azoles. We recently isolated a strain (TIMM20092) from a patient insensitive to standard treatment with terbinafine or azoles. A mutation of the squalene epoxidase, which is targeted by terbinafine, has been detected in this strain, while resistance to azoles could be attributed to drug efflux pumps (Monod, M. et al. manuscript in preparation). Although still exceptional in onychomycosis in Europe, resistance to azoles seems to be common in India in *Trichophyton mentagrophytes* isolated from skin dermatophytosis [42,49].

### 6.3. Insensitivity of NDM Onychomycosis to Standard Treatments Used for T. unguium

Fungal cultures performed in many cases of onychomycosis, insensitive to conventional oral treatments with terbinafine and/or itraconazole, do not reveal the presence of *T. rubrum* or *T. interdigitale*, and treatment failure cannot be attributed to acquired resistance. Pan-fungal PCR using DNA extracted from nail samples showed that an NDM (*Fusarium* sp., *Acremonium* sp. *or Aspergillus* sp.) was the infectious fungus in such cases [50,51]. Numerous reports based on the identification of the fungus in culture have shown that onychomycoses due to *Fusarium* sp. [52,53,54,55] are difficult to treat. *Fusarium* sp. generally affects the nail of the big toe (hallux), especially in patients who have suffered trauma or repetitive injuries. A mould must be suspected of being the infectious agent of onychomycosis upon failure of several treatments, with a positive direct examination without dermatophytes in culture, and a hallux infection without any other sign of inter-toe, plantar or anamnestic mycotic fungus of the foot.

For *Fusarium* sp., *Acremonium* sp. and *Aspergillus* sp. onychomycosis, a topical treatment is prescribed consisting of daily application of an amphotericin B solution (2.0 mg/mL in a 1:1 mixture of DMSO and isopropyl alcohol) [51]. Amphotericin B is a broad-spectrum antifungal agent of the polyene class and is fungicidal. This antifungal compound forms channels in the cell membrane of the fungus that allow ions and organic compounds of the cytoplasm to escape, thereby destroying membrane function [56]. In a pilot trial, treatment was effective in all patients (n = 8) [51]. Nails showed a healthy appearance after 6–12 months of solution application, and direct mycological examination was negative. The application of amphotericin B as a topical solution is a safe and relatively inexpensive treatment of choice for onychomycosis caused by moulds (Figure 2) [51,57].

## 7. Conclusions

While direct examination remains essential in medical mycology, a positive direct examination does not automatically mean that the infecting fungus is a dermatophyte. The identification of the infectious agent in onychomycosis, at least at the genus level, is very important for appropriate treatment prescription, before any investigation regarding possible acquired resistance by a dermatophyte. Cases of dermatophyte resistance to antifungal agents in nails are still rare (about 1%), and most cases of onychomycosis insensitive to standard treatments with terbinafine and azoles are due to mould infection.

In many laboratories, direct mycological examination and culture are performed routinely for each suspected case of onychomycosis. The development of PCR methods is underway to automate the analysis of onychomycosis samples. However, automation could be problematic because the material collected varies from a few nail scrapings to the entire nail. For example, 830 samples received out of 2267 could not be retained due to a lack of collected material in a study where real-time PCR results were compared to cultures [58]. The high rate of negative PCR results was on the order of 50% but the samples were not selected by direct mycological examination. We observed in our laboratory that the number of negative PCR results was greatly reduced by analysing only samples that are positive by direct mycological examination. Direct mycological examination seems difficult to eliminate in the laboratory because fungal elements can be detected in very little material.

With monoclonal antibodies, the dermatophyte strip tests for rapid detection of *Trichophyton* by immunochromatography could be appropriate in private practice. The tests developed in Japan have both high sensitivity and negative predictive value and are suitable for ruling out onychomycosis in doubtful cases [14,15,16,17,18]. Research on infected nails is technically demanding, but new knowledge of onychomycosis has proven to be useful in improving diagnosis and optimal therapeutic prescription.

## Figures and Tables

**Figure 1 jof-05-00020-f001:**
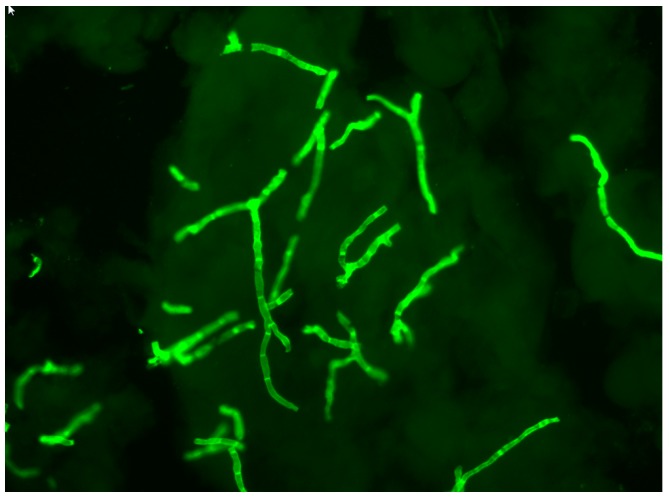
Direct mycological examinations using fluorescence microscopy showing hyphae in an infected nail. Bar: 20 μm.

**Figure 2 jof-05-00020-f002:**
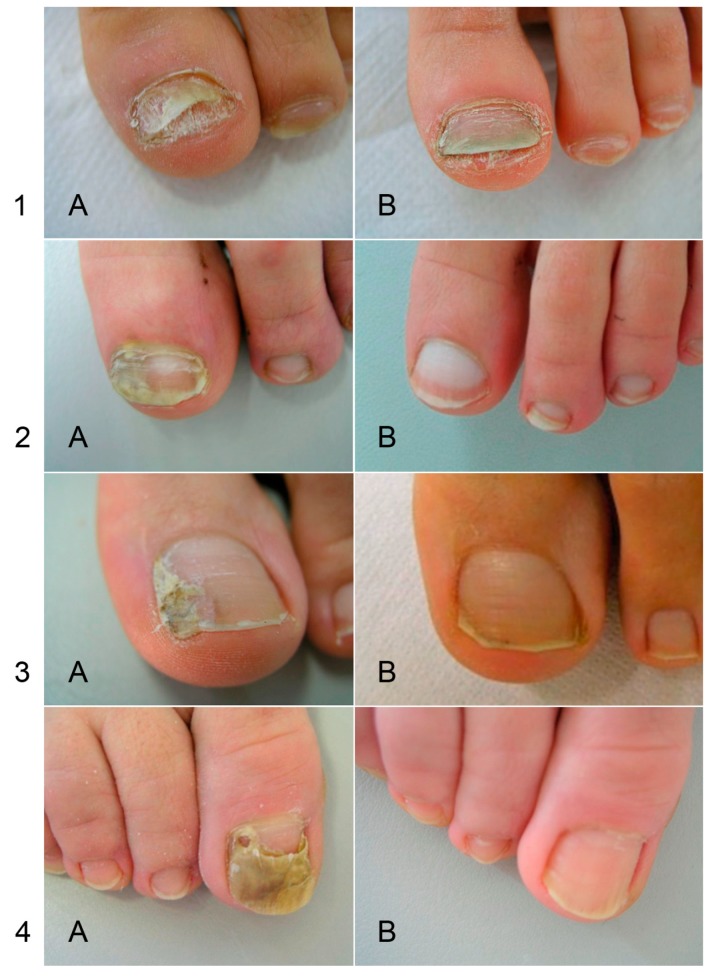
Efficacy of treatment with amphotericin B in topical application for NDM onychomycoses. (**A**) before treatment; (**B**) after treatment. The infectious agent identified by PCR was *Fusarium* sp. in patients 1 and 2, and *Aspergillus oryzae* in patients 3 and 4. Figure from Monod et al., Rev Med Suisse. 2013, 380:730-3, with permission of the Editor.

**Table 1 jof-05-00020-t001:** Tools for the diagnosis of onychomycosis and rapid identification of the pathogen.

Methods	Target	Usage §	Time	Specificity #	References
**Direct Mycological Examination**	Hyphae and spores in nails	PP, RL	2–5 min	Fungi (*Trichophyton*, NDMs, Yeasts)	[2,3,4,5]
**Cultures**	Growing fungi	RL	7–14 days	Tru, Tint NDMs if repeated isolations Yeasts if repeated isolations	[6]
**PCR with Specific Primers**	Nail fungal DNA	RL	5 h	Tru	[7,8,9]
**PCR with Pan-Fungal Primers /Sequencing**	Nail fungal DNA	RL	24 h	*Trichophyton* NDMs Yeasts	[10,11]
**Mass Spectrometry (MRM)**	Sub6	RD		Tru, Tint	[12]
**Western Blotting**	Sub6	RD	24 h	*Trichophyton*	[13]
**ELISA**	Sub6	ND	24 h	*Trichophyton*	[13]
**Strip Tests**	Polysaccharides	PP	15 min	*Trichophyton*	[14,15,16,17,18]

§ PP: usage in private practice; RL: usage in routine laboratory; RD: for research and development. # NDM: non-dermatophyte filamentous mould; Tru: *T. rubrum*; Tint: *T. interdigitale.*

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
