# Peer review of "Recent Findings in Onychomycosis and Their Application for Appropriate Treatment"

_jof, 2019, doi:10.3390/jof5010020_

Reviewer 1 Report

This paper on onychomycosis, while not strictly speaking a review is interesting and refreshing largely because it has adopted a very different approach to others that have discussed this field. It focuses on pathogenesis, diagnosis through introducing tests infrequently discussed such as antigen detection and therapy ; again the latter focuses  on two aspects that have not been well described such as  genetic evidence of terbinafine resistance and the treatment of mould infections. I felt that to be complete the authors should try to comment on these loose ends. 

How do dermatophytes and even more so non- dermatophytes invade nail?

Is there any other test that would eliminate the need for direct microscopy in an increasingly molecular diagnostic world ?

What about azole resistance and failure of nail treatment – after all in the Indian epidemic of drug failures in dermatophytosis itraconazole is the main drug where treatment failure is common, although few cases have onychomycosis.

Overall the authors need to revise the English in places as it is not always clear and there are some unusual words eg antifongigrams. Also check manuscript eg define TTT

Author Response

Responses to comments of the reviewers

Reviewer 1

This paper on onychomycosis, while not strictly speaking a review is interesting and refreshing largely because it has adopted a very different approach to others that have discussed this field. It focuses on pathogenesis, diagnosis through introducing tests infrequently discussed such as antigen detection and therapy ; again the latter focuses  on two aspects that have not been well described such as  genetic evidence of terbinafine resistance and the treatment of mould infections. I felt that to be complete the authors should try to comment on these loose ends. 

How do dermatophytes and even more so non- dermatophytes invade nail?

Response

The following paragraph was added, page 7

The fungus can enter the nail through the distal subungual area and the lateral nail groove, through the dorsal surface of the nail plate producing superficial onychomycosis or through the under-surface of the proximal fold of the nail [1]. Infection may also be secondary to paronychia. Dermatophyte infections are often the result of an intertrigo, while repetitive strain injuries to the nail, especially hallux, may play a major role in mold invasion.

Is there any other test that would eliminate the need for direct microscopy in an increasingly molecular diagnostic world ?

Response

The introduction of an "all PCR" procedure in a laboratory is tempting, but such a procedure is difficult in practice because of the heterogeneity of the nail samples. For example, 830 samples received out of 2267 could not be retained due to a lack of collected material in a study where real-time PCR results were compared to cultures [57]. The high rate of negative PCR results was in the order of 50% but the samples were not selected by direct mycological examination. We observed in our laboratory that the number of negative PCR results was greatly reduced by analyzing only samples that are positive by direct mycological examination. Direct mycological examination seems difficult to eliminate in the laboratory because fungal elements can be detected in very little material.

A complement was added in last paragraph “Conclusion”

What about azole resistance and failure of nail treatment – after all in the Indian epidemic of drug failures in dermatophytosis itraconazole is the main drug where treatment failure is common, although few cases have onychomycosis.

Response

The following sentence with two new references was added at page 12

Although still exceptional in onychomycosis in Europe, resistance to azoles seems to be common in India in Trichophyton mentagrophytes from skin dermatophytosis [49-50].

Overall the authors need to revise the English in places as it is not always clear and there are some unusual words eg antifongigrams. Also check manuscript eg define TTT

Response

“Antifongigram” was changed by “Antifungal susceptibility testing”

TTT ” was changed by “treatment”

Reviewer 2

Although this is a laboratory focused work, it is of interest to any clinician who may treat onychomycosis, and not only dermatologists. For this reason I would change the word "dermatologist" wherever it appears to "clinician."

Response

We changed "dermatologist" by "clinician" all through the ms.

The sentence "There is no mycosis....and 50% of onychodystrophy cases are not due to fungi" on lines 43-44 is a very often cited stamen so it should be referenced in my opinion.

Response

We agree with this comment. In fact, this sentence was not necessary. Therefore, it was deleted.

On line 166, "Terbinafine and azole are".... should be changed to "Terbinafine is used" . This is because on line 166 you then describe how azoles act. The way it is written now the first line seems to indicate that terbinafine and azoles have the same mechanism of action.

Response

The ms was modified accordingly as following (see page 10):

“Terbinafine and azole are used extensively in therapy for dermatophyte infections. Terbinafine inhibits the squalene epoxidase (SQLE) involved in an early step of ergosterol biosynthesis, [36] a compound essential for the structure and function of the plasma membrane in fungi. …”

Reviewer 2 Report

Although this is a laboratory focused work, it is of interest to any clinician who may treat onychomycosis, and not only dermatologists. For this reason I would change the word "dermatologist" wherever it appears to "clinician."

The sentence "There is no mycosis....and 50% of onychodystrophy cases are not due to fungi" on lines 43-44 is a very often cited stamen so it should be referenced in my opinion.

On line 166, "Terbinafine and azole are".... should be changed to "Terbinafine is used" . This is because on line 166 you then describe how azoles act. The way it is written now the first line seems to indicate that terbinafine and azoles have the same mechanism of action.

Author Response

Responses to comments of the reviewers

Reviewer 2

Although this is a laboratory focused work, it is of interest to any clinician who may treat onychomycosis, and not only dermatologists. For this reason I would change the word "dermatologist" wherever it appears to "clinician."

Response

We changed "dermatologist" by "clinician" all through the ms.

The sentence "There is no mycosis....and 50% of onychodystrophy cases are not due to fungi" on lines 43-44 is a very often cited stamen so it should be referenced in my opinion.

Response

We agree with this comment. In fact, this sentence was not necessary. Therefore, it was deleted.

On line 166, "Terbinafine and azole are".... should be changed to "Terbinafine is used" . This is because on line 166 you then describe how azoles act. The way it is written now the first line seems to indicate that terbinafine and azoles have the same mechanism of action.

Response

The ms was modified accordingly as following (see page 10):

“Terbinafine and azole are used extensively in therapy for dermatophyte infections. Terbinafine inhibits the squalene epoxidase (SQLE) involved in an early step of ergosterol biosynthesis, [36] a compound essential for the structure and function of the plasma membrane in fungi. …”